# Evaluating the higher-order structure of the Profile of Emotional Competence (PEC): Confirmatory factor analysis and Bayesian structural equation modeling

Yuki Nozaki[1]*, Alicia Puente-Martínez[2], Moïra Mikolajczak[3]

**1** Department of Human Science, Faculty of Letters, Konan University, Kobe, Japan, **2** Department of Social Psychology and Methodology of Behavior Sciences, University of the Basque Country, Lejona, Spain, **3** Department of Psychology, Université catholique de Louvain, Louvain-la-Neuve, Belgium

* y_nozaki@konan-u.ac.jp

**Data Availability Statement:** All data and Mplus syntaxes needed for analyses are available from the Open Science Framework (https://osf.io/mwpxt/).

## Abstract

Emotional competence (EC) reflects individual differences in the identification, comprehension, expression, regulation, and utilization of one's own and others' emotions. EC can be operationalized using the Profile of Emotional Competence (PEC). This scale measures each of the five core emotional competences (identification, comprehension, expression, regulation, and utilization), separately for one's own and others' emotions. However, the higher-order structure of the PEC has not yet been systematically examined. This study aimed to fill this gap using four different samples (French-speaking Belgian, Dutch-speaking Belgian, Spanish, and Japanese). Confirmatory factor analyses and Bayesian structural equation modeling revealed that a structure with two second-order factors (intrapersonal and interpersonal EC) and with residual correlations among the types of competence (identification, comprehension, expression, regulation, and utilization) fitted the data better than alternative models. The findings emphasize the importance of distinguishing between intrapersonal and interpersonal domains in EC, offer a better framework for differentiating among individuals with different EC profiles, and provide exciting perspectives for future research.

## Introduction

Individuals differ in the extent to which they can appropriately identify, understand, express, regulate, and utilize their own and others' emotions. The concept of "emotional competence" (EC)––alternatively labeled "emotional intelligence" (EI)––has been proposed to account for this idea. Although the term EC was originally proposed to account for these individual differences [1], the term EI was later proposed and became much more popular. However, we prefer the term EC to EI because recent meta-analysis shows that they can be improved via relatively short trainings, unlike intelligence [2]. Given this line of research, we will use the term EC hereafter as a synonym of EI, in accordance with previous research [3–8].

**Funding:** This research was supported by JSPS KAKENHI (grant number: 17H06780) to YN, Spanish Ministry of Science and Innovation (grant number: PSI2014-51923-P; PSI2017-84145-P) to AP-M, and University of the Basque Country and Basque Government (grant number: GIC12/91 IT-666-13) to AP-M. The funders had no role in study design, data collection and analysis, decision to publish, or preparation of the manuscript.

**Competing interests:** The authors have declared that no competing interests exist.

Whether called EC or EI, the nature of these emotion-related differences has long been a source of debate among researchers. Some authors view them as the result of differences in abilities [9], others personality [10] and still others as the result of a mix of both [11]. The tripartite model proposed by Mikolajczak, Petrides [12] integrates these different conceptions by considering that people can difference in emotion-related knowledge, abilities and traits. The knowledge level refers to what people know about emotions and emotionally competent behaviors (e.g., Do I know which emotional expressions are constructive in a given social situation?). The ability level refers to the ability to apply this knowledge in a real-world situation (e.g., Am I able to express my emotions constructively in a given social situation?). The trait level refers to emotion-related dispositions, namely, the propensity to behave in a certain way in emotional situations (e.g., Do I typically express my emotions in a constructive manner in social situations?). These three levels of emotion-related individual differences are loosely connected [13]. In the current paper, we focus on the trait level typically measured through self-report questionnaires [14] because the trait-level is more strongly associated with adjustment than the ability-level is [15–19].

Previous research has shown that the trait level of EI/EC is positively associated with better adjustment, such as more adaptive emotion regulation [20], greater subjective well-being [18], better mental and psychical health [16, 21], higher academic performance [22], higher job satisfaction [23, 24], less counterproductive work behavior [17] and greater romantic relationship satisfaction [25]. These relationships remain significant after controlling for personality or cognitive ability [26, 27].

To assess the trait-level EC, Brasseur, Gregoire [28] recently developed the Profile of Emotional Competence (PEC). This scale assesses 10 core EC facets: five types of competences (emotion identification, emotion comprehension, emotion expression, emotion regulation, and emotion utilization), each comprising an intrapersonal domain (concerning one's own emotions) and an interpersonal domain (concerning others' emotions). These five competences derive from the four-branch model proposed by Mayer and Salovey [9]; however, they separate the identification from the expression of emotions based on research on alexithymia showing that these branches are factorially and conceptually distinct [29]. A strength of the PEC is that it can assess both intrapersonal and interpersonal domains in all five core competences. Moreover, previous research has found that it had an adequate reliability and incremental validity over the Big Five personality traits [6, 28]. Given its strengths, the PEC has been rapidly adopted in recent EC research [5, 7, 30–37].

Because EC facets are positively related to each other [28], they will be hierarchically structured. Clarification of the higher-order structure of individual differences is important because it can provide a parsimonious summary of the vast complexity of human nature [38]. Given that the above 10 core EC facets are categorized into a 2 (type of target) × 5 (type of competence) framework, we can assume six possible structures. These six candidate models are depicted in Fig 1 and briefly described hereafter.

Unidimensional structure: The core 10 EC facets form only one higher-order factor (global EC). This model will serve as a baseline for model comparison in the statistical analyses.

Target-based structure: The 10 core EC facets form two higher-order factors: intrapersonal and interpersonal EC. These factors do not distinguish between the type of competence (emotion identification, emotion comprehension, emotion expression, emotion regulation, or emotion utilization).

Competence-based structure: The 10 core EC facets form five higher-order factors (emotion identification, emotion comprehension, emotion expression, emotion regulation, and emotion utilization) that do not distinguish between intrapersonal and interpersonal competence.

### Unidimensional structure

### Target-based structure

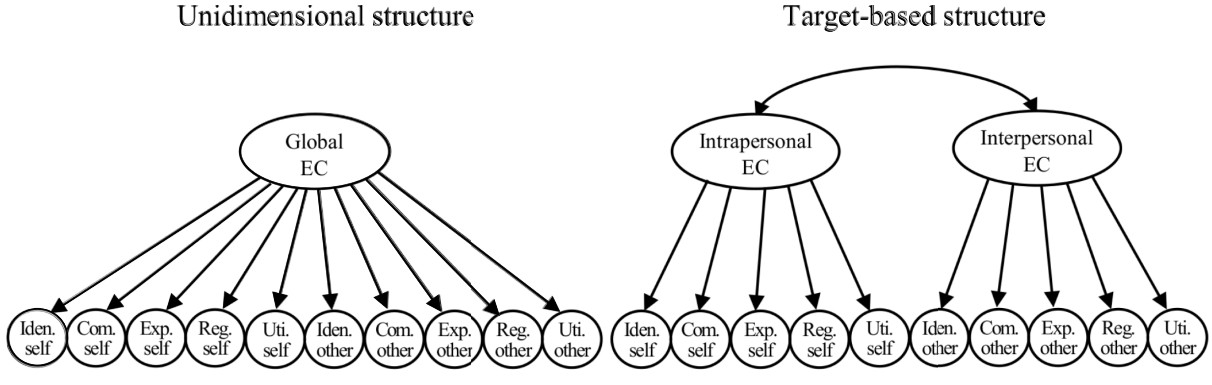

### Competence-based structure

### Hybrid structure

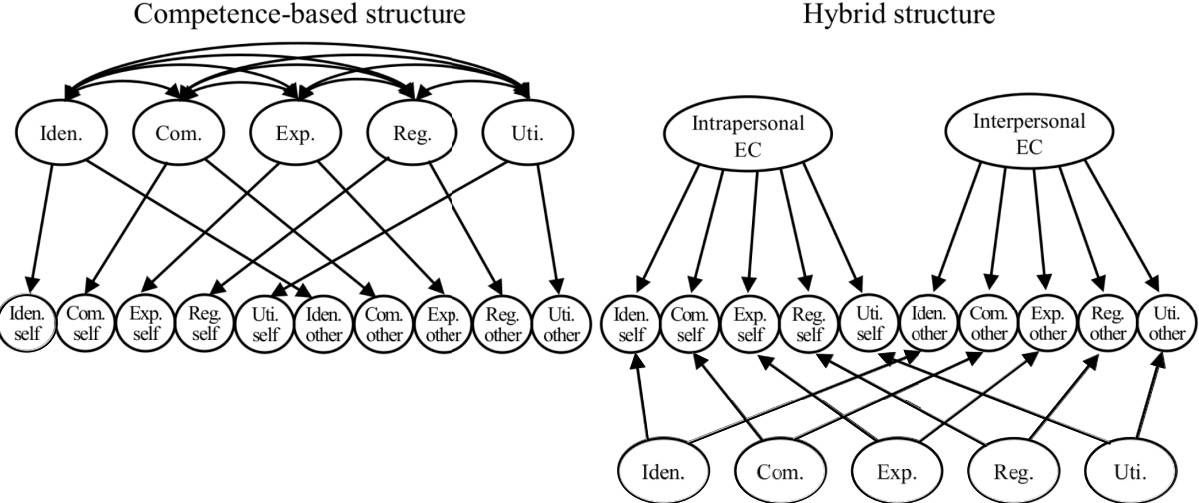

### Modified target-based structure

### Modified competence-based structure

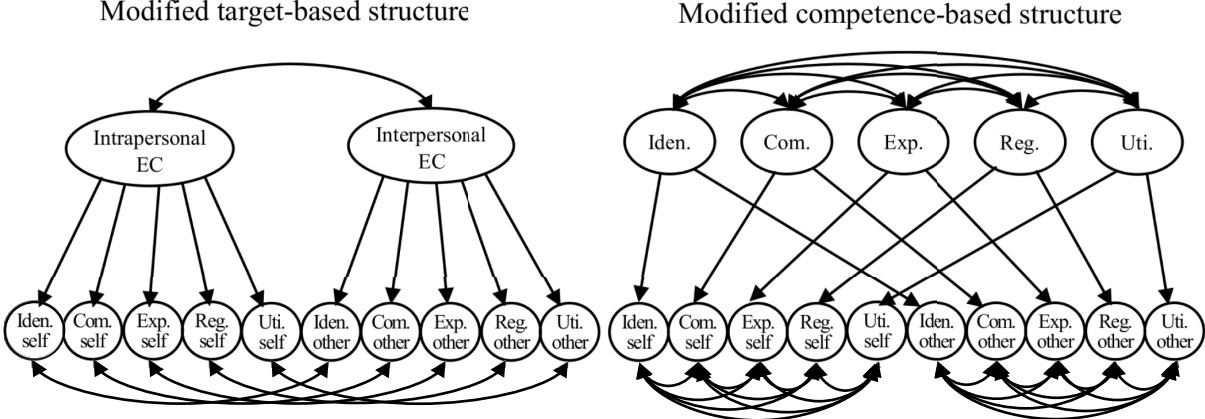

**Fig 1. Candidate factorial models for emotional competence.** EC: emotional competence, Iden.: emotion identification, Com.: emotion comprehension, Exp.: emotion expression, Reg.: emotion regulation, Uti.: emotion utilization.

Hybrid structure: Instead of a normal second-order factor model, we can use a hybrid model [39, 40]—an extension to the bifactor model to capture the 2 (type of target) × 5 (type of competence) crossed structure. The 10 core EC facets form two types of dimensions (type of target and type of competence) to yield additive effects. We provide further details and previous research applications of this model in S1 Text.

Modified target-based structure and modified competence-based structure: In the hybrid structure model, if factors are unstable, they can be replaced with residual correlations [41, 42]. Thus, we can also test a model replacing factors of competence-based structure with residual correlations in the hybrid structure (i.e., a modified target-based structure) or replacing factors of target-based structure with residual correlations in the hybrid structure (i.e., a modified competence-based structure).

The authors of the PEC [28] originally assumed 10 first-order factors and two second-order factors (intrapersonal and interpersonal EC), as corroborated by other research [6, 32]. However, to the best of our knowledge, no study has ever systematically compared this target-based structure with other theoretically plausible factor structures. Consequently, the optimal model for the PEC is still unknown. To fill this gap, we compared the fit of the six theoretically plausible models and tested the replicability/stability of the results across four different samples (French-speaking Belgian, Dutch-speaking Belgian, Spanish, and Japanese).

To evaluate the best factor structure, we followed the flowchart recently proposed by Schmitt, Sass [38]. They encourage researchers to start by conducting dimensionality analyses (e.g., parallel analysis, minimum average partial test, exploratory graph analysis); then, if theoretical candidate factor structures exist, they recommend confirmatory factor analysis (CFA). After that, if model fit is not sufficient, they recommend Bayesian structural equation modeling (BSEM) to explore the source of model misfit.

Previous research has emphasized that the model constraints in traditional CFA are unrealistic for the study of hierarchical constructs. For example, Hopwood and Donnellan [43] found that widely used personality trait inventories (e.g., the Revised NEO Personality Inventory [44]) usually demonstrate poor model fit when their structure is evaluated with CFA. This failure is due to the inherent complexity of hierarchical constructs: In typical CFA, cross-loadings and residual correlations are presumed fixed at exact zero, but these unnecessarily strict models lead to poor model fit and substantial parameter biases for factor loadings and correlations [45, 46]. Nevertheless, freer parameters for cross-loadings and residual correlations would result in a non-identified model under the traditional CFA.

To solve this issue, Muthén and Asparouhov [46] proposed a new statistical approach, called BSEM. This approach allows simultaneous estimation of all cross-loadings and residual correlations by using approximate zero informative priors to replace the exact zeros for those loadings and correlations. By applying BSEM, researchers can investigate whether model misfit is due to small or large cross-loadings/residual correlations, missing factors, or extra factors [41]. BSEM has already been successfully applied to various existing cognitive and non-cognitive measures [41, 46–52]. Thus, we apply BSEM to investigate source of model misfit if the fit of the best traditional CFA model is not sufficient.

## The current research

This study aimed to evaluate the higher-order structure of the PEC using Schmitt, Sass [38]'s guidelines. As recommended in their flowchart, we started with dimensionality analyses, followed by traditional CFA and BSEM. In order to test the stability and replicability of the results, we evaluated the structure of PEC across four different language samples from Western and Eastern cultures (French-speaking Belgian, Dutch-speaking Belgian, Spanish, and Japanese).

## Method

### Participants and procedure

*Sample A* consisted of 3295 French-speaking Belgians (males = 1355, females = 1854, unanswered = 86, *M*age = 53.36, *SD* = 14.01), who completed the French version of the PEC. *Sample B* consisted of 9955 Dutch-speaking Belgians (male = 3746, female = 5850, unanswered = 359, *M*age = 55.62, *SD* = 13.34), who completed the Dutch version of the PEC. Sample A and B were derived from a part of a study conducted by the largest Mutual Benefit Society in Belgium. The data have already been used to answer other research questions (i.e., on the impact of EC on healthcare service use; [30, 37]); however, no factor analysis of EC has ever been conducted on these data. *Sample C* consisted of 792 Spanish people (male = 278, female = 512, unanswered = 2, *M*age = 24.07, *SD* = 8.44), who completed the Spanish version of the PEC over the course of a university semester. The survey was conducted using SurveyMonkey, and sent via email to all students enrolled in the course. *Sample D* consisted of 549 Japanese people (male = 344, female = 205, *M*age = 31.67, *SD* = 14.45), who completed the Japanese version of the PEC. They were recruited via a Japanese data collection company (Cross Marketing Inc.)

Participants in all samples answered the questionnaire online. At the beginning of the survey, they were informed about the nature of the study, including the study's purpose, their right to withdraw from the study, and the confidentiality of their responses. After reading this material, participants provided informed consent by clicking the "accept" button to start the survey. In addition to the EC scale, participants completed other measures unrelated to the present research question. This study was approved by the ethics committees of the Université catholique de Louvain, University of the Basque Country, and Kyoto University.

### Measure

EC was assessed with the PEC [28]. This scale comprises 10 first-order subscales with five items each: identification-self (e.g., I am aware of my emotions as soon as they arise), comprehension-self (e.g., As my emotions arise, I don't understand where they come from; reversed item), expression-self (e.g., I am good at describing my feelings), regulation-self (e.g., When I am sad, I find it easy to cheer myself up), utilization-self (e.g., My emotions inform me about changes I should make in my life), identification-other (e.g., I can tell whether a person is angry, sad or happy even if they don't talk to me), comprehension-other (e.g., Most of the time I understand why people feel the way they do), expression-other (e.g., I find it difficult to listen to people who are complaining; reversed item), regulation-other (e.g., I am good at lifting other people's spirits), and utilization-other (e.g., If I wanted, I could easily influence other people's emotions to achieve what I want).

All translated measures were created via a back-translation procedure. Participants in samples A, B and D rated each item on a 5-point scale, whereas, participants in sample C rated it on a 7-point scale, because this sample were relatively homogeneous (i.e., everyone was a student). To increase the potential to detect true variation, the number of response options was increased [53]. Importantly, this modification did not affect our main results, because we found similar factor structure across all samples, as described in the results section.

### Statistical analyses

First, we conducted dimensionality analyses based on the first-order facet scores, using the exploratory graph analysis [54]. This method has been shown to be superior to other traditional dimensionality analysis methods such as the parallel analysis or the minimum average

partial test [54, 55]. Exploratory graph analysis with a triangulated maximally filtered graph was conducted using the EGA 0.4 package [56] in R 3.5.0 [57].

Next, we implemented CFA to compare the fit of possible factor structure models (Fig 1). All CFA were conducted with Mplus Version 8.2 [58]. Since the normalized estimate of Mardia's coefficient indicated that multivariate normality was violated, we applied a robust maximum likelihood (MLR) estimator and the Satorra–Bentler scaled $\chi^2$. There is a controversy as to whether MLR or weighted least squares mean- and variance-adjusted (WLSMV) estimation is superior when multivariate normality is violated [59]. However, neither the Akaike Information Criterion (AIC) nor the Bayesian information criterion (BIC) can be computed with WLSMV, while both can with MLR. Because AIC and BIC are frequently used for model comparison, we used MLR in this study. To help parameter estimation, we constrained paths from the same second-order factors with only two indicators (i.e., competence-based structure, hybrid structure, and modified competence-based structure), as in previous studies [60]. We used AIC and BIC for model comparison; lower BIC and AIC suggest better model fit. Moreover, we used the comparative fit index (CFI; a value $\geq$ .90 suggests acceptable fit), standardized root mean square residual (SRMR; a value $\leq$ .08 suggests acceptable fit), and root mean square error of approximation (RMSEA; a value $\leq$ .08 suggests acceptable fit) to evaluate overall model fit [61, 62]. Missing values (only 0.013%) were handled by full information maximum likelihood estimation (software default settings).

If the fit indices of the best-selected model are not sufficient in CFA with MLR, Schmitt, Sass [38] recommend BSEM to explore source of model misfit. Here, the BSEM models were estimated using the Bayes estimator with a series of prior specifications for cross-loadings and residual correlations with the standardized item scores. All BSEM were conducted with Mplus Version 8.2 [58]. For metrics, we fixed one relatively stable first-order factor loading per factor and set variances of second-order factors at one. First, BSEM models specified noninformative priors for the hypothesized factor loadings, but did not estimated cross-loadings and residual correlations. Next, we specified small-variance informative priors for the cross-loadings, choosing normal prior distributions $N$ (0, 0.01) yielding 95% small cross-loading bounds of $\pm$0.20 [46]. Finally, we added informative Inverse Wishart ($dD,d$) priors for the residual variances/covariances [41], where $D$ refers to the residual variance/covariance of the Bayesian CFA models and $d$ refers to the degrees of freedom. We used $d$ = 1000 as a starting value; then, we conducted the sequence of sensitivity analyses described in Asparouhov, Muthén [41]. If convergence was fast but model fit was unacceptable (PP$p$ < .05), the next step reduced $d$ (e.g., -100) and repeated the analyses. If slow or no convergence happened, the next step increased $d$ (e.g., + 100) and again repeated the analyses. This sensitivity analysis procedure was intended to change the variance of the small priors to monitor the distance between the data and the model. As explained in Asparouhov and Muthén [63], "In this process no particular prior variance is preferred, rather, the prior variance is adjusted gradually to maintain identifiability of the model while resolving model fit and separating parameters that have minor deviations from zero from substantively important misspecifications" (p. 2).

The BSEM estimation was run with three independent Markov chain Monte Carlo chains using the Gibbs sampler [41, 46], with 150,000 iterations (of which the first 75,000 were discarded as the burn-in phase). No thinning was conducted. Model convergence was monitored by potential scale reduction (a value $\leq$ 1.10 suggests convergence) and visually checking trace plots. Model fit was evaluated using the posterior predictive $p$-value (PP$p$) with associated 95% confidence interval; a PP$p$ < .05 and a positive 95% lower limit imply a poor model fit. The deviance information criterion (DIC) was used for comparison of BSEM models because it is more appropriate than BIC for BSEM [41]; lower DIC suggests better model fit. Moreover, when we used approximately zero priors for cross-loadings and/or residual correlations, prior-

posterior predictive $p$-value (PPP$p$) was used to test for the hypothesis that a set of parameters are approximately zero [63, 64]; a PPP$p < .05$ imply that this hypothesis is rejected. All data and Mplus syntaxes needed for analyses are available at https://osf.io/mwpxt/.

## Results

### Dimensionality analyses and CFA with MLR

Exploratory graph analysis showed that two common factors were recommended in all samples. Next, we conducted CFA with MLR to compare model fit of possible factor structures. Fit indices of each model are shown in Table 1. In all samples, AIC and BIC were lower for the target-based structure than for the unidimensional EC structure. Moreover, an improper solution (the psi matrix is not positive definite) was found for the competence-based structure. This improper solution emerged because some correlation coefficients among second-order factors

**Table 1. Fit indices of CFA with a robust maximum likelihood estimation.**

| Model | S-B $\chi^2$ | df | CFI | SRMR | RMSEA [90%CI] | AIC | BIC |
|---|---|---|---|---|---|---|---|
| **Sample A: French-speaking Belgian ($n$ = 3295)** | | | | | | | |
| I. Unidimensional structure model | 11903.36*** | 1165 | .752 | .074 | .053 [.052, .054] | 432663.82 | 433639.84 |
| II. Target-based structure model | 11072.80*** | 1164 | .771 | .071 | .051 [.050, .052] | 431662.44 | 432644.57 |
| III. Competence-based structure model | Improper solution (the psi matrix is not positive definite)[a] | | | | | | |
| IV. Hybrid structure model[b] | 12002.03*** | 1160 | .749 | .128 | .053 [.052, .054] | 432790.38 | 433796.91 |
| V. Modified target-based structure model | 10833.01*** | 1159 | .776 | .071 | .050 [.049, .051] | 431378.87 | 432391.49 |
| VI. Modified competence-based structure model | Improper solution (the psi matrix is not positive definite)[a] | | | | | | |
| **Sample B: Dutch-speaking Belgian ($n$ = 9955)** | | | | | | | |
| I. Unidimensional structure model | 32717.74*** | 1165 | .741 | .075 | .052 [.052, .053] | 1240702.91 | 1241855.84 |
| II. Target-based structure model | 30814.93*** | 1164 | .757 | .073 | .051 [.050, .051] | 1238366.84 | 1239526.98 |
| III. Competence-based structure model | Improper solution (the psi matrix is not positive definite)[a] | | | | | | |
| IV. Hybrid structure model[b] | 34435.04*** | 1160 | .727 | .134 | .054 [.053, .054] | 1242779.92 | 1243968.88 |
| V. Modified target-based structure model | 30195.92*** | 1159 | .762 | .074 | .050 [.050, .051] | 1237618.45 | 1238814.61 |
| VI. Modified competence-based structure model | Improper solution (the psi matrix is not positive definite)[a] | | | | | | |
| **Sample C: Spanish ($n$ = 792)** | | | | | | | |
| I. Unidimensional structure model | 5322.47*** | 1165 | .645 | .103 | .067 [.065, .069] | 135376.74 | 136124.67 |
| II. Target-based structure model | 5137.93*** | 1164 | .660 | .100 | .066 [.064, .067] | 135142.27 | 135894.87 |
| III. Competence-based structure model | Improper solution (the psi matrix is not positive definite)[a] | | | | | | |
| IV. Hybrid structure model[b] | 5296.40*** | 1160 | .646 | .139 | .067 [.065, .069] | 135366.28 | 136137.59 |
| V. Modified target-based structure model | 5085.29*** | 1159 | .664 | .100 | .065 [.064, .067] | 135085.45 | 135861.43 |
| VI. Modified competence-based structure model | Improper solution (the psi matrix is not positive definite)[a] | | | | | | |
| **Sample D: Japanese ($n$ = 549)** | | | | | | | |
| I. Unidimensional structure model | 3569.89*** | 1165 | .713 | .078 | .061 [.059, .064] | 71654.89 | 72344.18 |
| II. Target-based structure model | 3437.59*** | 1164 | .729 | .076 | .060 [.057, .062] | 71511.00 | 72204.60 |
| III. Competence-based structure model | Improper solution (the psi matrix is not positive definite)[a] | | | | | | |
| IV. Hybrid structure model[b] | 3605.94*** | 1160 | .708 | .142 | .062 [.060, .064] | 71728.47 | 72439.302 |
| V. Modified target-based structure model | 3412.66*** | 1159 | .731 | .075 | .060 [.057, .062] | 71481.33 | 72196.478 |
| VI. Modified competence-based structure model | Improper solution (the psi matrix is not positive definite)[a] | | | | | | |

*Note*. CFA: confirmatory factor analysis, S-B $\chi^2$: Satorra-Bentler scaled $\chi^2$

[a] Some correlation coefficients among second-order factors exceeded 1.00, suggesting factors were overextracted.

[b] Rindskopf (1983)'s reparameterization was applied.

***$p < .001$

(e.g., correlation between emotion identification and emotion expression) exceeded 1.00, implying factors were overextracted (for detailed factor loadings, see S1 Table). These results suggest that target-based structure is superior to the unidimensional structure and the competence-based structure.

For the hybrid structure, some variances were negative, suggesting an improper solution. Lance and Fan [65] indicated that this improper solution usually happens in a hybrid-structure model. To solve this issue, they recommended Rindskopf [66]'s reparameterization, which fixes the variance of the residual at one and estimates the coefficient. Following their recommendation, we applied Rindskopf [66]'s reparameterization to the hybrid model; it returned proper solutions in all samples. Although the model fit of the hybrid structure was inferior to that of the target-based structure, the patterns of second-order factor loadings were interesting: factor loadings from the target-based structure (intrapersonal and interpersonal EC, average factor loadings = .75) were much stronger than those from the competence-based structure (emotion identification, emotion comprehension, emotion expression, emotion regulation, and emotion utilization; average factor loadings = .21; for detailed factor loadings, see S2 Table).

With regards to the modified target-based structure, where competence factors in the hybrid structure were replaced by residual correlations, AIC and BIC were the lowest among the possible models, in all samples. Moreover, as in the competence-based structure, an improper solution (non-positive-definite psi matrix) was found for the modified competence-based structure in all samples, because some correlation coefficients among second-order factors exceeded 1.00, implying that factors were overextracted. Taken together, these results suggest that the modified target-based structure is best to represent the EC factor structure as assessed with the PEC.

Standardized second-order factor loadings deriving from the modified target-based structure are shown in Table 2. All hypothesized major loadings were substantially large ($\geq$ .36) and statistically significant. Moreover, when looking at residual correlations among first-order factors, correlations between *regulation-self* and *regulation-other* were substantially large in all samples (*r*s = .41 to .55). However, although SRMRs (except for sample C) and RMSEAs showed adequate fit, CFIs were not acceptable in all samples even for the best-fitted modified two-second-order-factor model. Therefore, we explored the source of model misfit using BSEM.

## BSEM

We conducted BSEM using the modified target-based structure model. Table 3 presents the fit indices of the results. In all samples, BSEM with no informative priors and BSEM with cross-loadings were rejected by the data (PP*p* $\leq$ .001), with a high 95% lower PP limit. Therefore, we added informative priors for the residual variances/covariances. When *d* was set to 1000, BSEM analyses gave PP*p* values higher than .05 in sample B (.278), but lower than .05 in samples A, C, and D (PP*p* $\leq$ .042). Therefore, the next step decreased *d* by 100 and repeated the analyses with the new *d*. This procedure was repeated until sufficient model fit was achieved. When *d* was set to 200, PP*p* values were greater than 0.05 in all samples (.206 to .660). Thus, we adopted *d* = 200 to maintain the identifiability of the model while resolving model fit and separating parameters that had minor deviations from zero from substantively important misspecifications.

Potential scale reductions were lower than 1.10 in all samples, and chains indicated clear mixing in trace plots, suggesting good convergence [46]. Following Depaoli and van de Schoot [67], we also checked whether convergence remained after doubling the number of iterations

**Table 2. Results of the CFA with a robust maximum likelihood estimation of the modified target-based structure model.**

| | Sample A: French-speaking Belgian (*n* = 3295) | | Sample B: Dutch-speaking Belgian (*n* = 9955) | | Sample C: Spanish (*n* = 792) | | Sample D: Japanese (*n* = 549) | |
|---|---|---|---|---|---|---|---|---|
| | Intrapersonal EC | Interpersonal EC | Intrapersonal EC | Interpersonal EC | Intrapersonal EC | Interpersonal EC | Intrapersonal EC | Interpersonal EC |
| Factor loadings | | | | | | | | |
| Identification-self | .95* [.92, .98] | | .96* [.95, .98] | | .98* [.92, 1.04] | | .95* [.89, 1.02] | |
| Comprehension-self | .87* [.84, .89] | | .85* [.84, .87] | | .79* [.71, .87] | | .84* [.75, .93] | |
| Expression-self | .81* [.78, .84] | | .83* [.81, .85] | | .78* [.69, .86] | | .83* [.75, .92] | |
| Regulation-self | .61* [.57, .65] | | .61* [.59, .64] | | .53* [.44, .63] | | .69* [.60, .78] | |
| Utilization-self | .36* [.30, .41] | | .37* [.32, .41] | | .44* [.33, .55] | | .39* [.24, .55] | |
| Identification-other | | .93* [.91, .95] | | .96* [.95, .97] | | .92* [.86, .99] | | .82* [.71, .93] |
| Comprehension-other | | .93* [.91, .96] | | .96* [.94, .97] | | .98* [.92, 1.04] | | .90* [.80, .99] |
| Expression-other | | .75* [.71, .78] | | .82* [.80, .84] | | .81* [.75, .87] | | .72* [.63, .81] |
| Regulation-other | | .86* [.82, .89] | | .83* [.81, .85] | | .76* [.68, .84] | | .89* [.77, 1.00] |
| Utilization-other | | .50* [.45, .55] | | .53* [.50, .56] | | .36* [.25, .47] | | .86* [.75, .97] |
| Factor correlation | | | | | | | | |
| Intrapersonal EC <-> Interpersonal EC | .71* [.68, .74] | | .77* [.75, .78] | | .67* [.59, .76] | | .73* [.63, .83] | |
| Residual correlations | | | | | | | | |
| Identification-self <-> Identification-other | .09 [-.15, .33] | | .01 [-.21, .22] | | -.48* [-1.69, .73] | | .21 [-.26, .67] | |
| Comprehension-self <-> Comprehension-other | .29* [.14, .45] | | .21* [.10, .31] | | -.27 [-.84, .31] | | .32* [.01, .63] | |
| Expression-self <-> Expression-other | .02 [-.07, .10] | | .07* [.01, .13] | | .03 [-.17, .23] | | .16 [-.20, .53] | |
| Regulation-self <-> Regulation-other | .55* [.47, .62] | | .51* [.47, .55] | | .46* [.35, .58] | | .41* [.20, .62] | |
| Utilization-self <-> Utilization-other | .11* [.05, .17] | | .12* [.08, .16] | | .12 [.01, .23] | | .20 [-.04, .43] | |

*Note.* 95% confidence intervals are in square brackets. EC: emotional competence. Although several upper bounds of 95% confidence intervals of standardized factor loadings were higher than one, this is normal and not a problem. For example, the results of Muthén and Asparouhov [46] also show that several upper bounds of 95% confidence intervals of standardized factor loadings were higher than one (see https://www.statmodel.com/BSEM.shtml for the their results on confidence intervals). *95% confidence interval does not include zero.

(300,000); potential scale reductions remained lower than 1.10 and deviations of parameters were $\leq |0.02|$ in all samples, suggesting good convergence. Moreover, PP*p* values were greater than 0.05 and the 95% PP limit did not include zero in all samples, suggesting good model fit. DIC showed that the model with cross-loadings and residual correlations was superior to the one with only cross-loadings and the one without cross-loadings or residual correlations, in all samples.

Standardized second-order factor loadings and factor correlations of this model (*d* = 200) are shown in Table 4 (for standardized first-order factor loadings and residual correlations, see S3 Table). All hypothesized major second-order factor loadings were substantively large ($\geq$ .34) and the credible interval did not include zero, except for the loading of *utilization-self* on intrapersonal EC (factor loadings = .14–.30). As in the results of CFA, intrapersonal and interpersonal EC were significantly correlated with each other (*r*s = .67–.80). Residual correlations between *regulation-self* and *regulation-other* were substantially large in all samples (*r*s = .39–.55).

**Table 3. Fit indices of Bayesian structural equation modeling of the modified target-based structure model.**

| Model | 2.5% PP limit | 97.5% PP limit | DIC | BIC | PP$p$ | PPP$p$ |
|---|---|---|---|---|---|---|
| Sample A: French-speaking Belgian ($n$ = 3295) | | | | | | |
| The model with no informative priors | 11689.52 | 11903.78 | 426830.90 | 427842.85 | .000 | – |
| The model with cross-loadings (prior variances = 0.1) | 1554.46 | 1809.31 | 417041.98 | 421126.38 | .000 | .000 |
| The model with cross-loadings (prior variances = 0.1) and residual correlations ($d$ = 200) | -171.09 | 111.28 | 416081.36 | 428903.03 | .660 | 1.00 |
| Sample B: Dutch-speaking Belgian ($n$ = 9955) | | | | | | |
| The model with no informative priors | 35481.30 | 35697.62 | 1297841.07 | 1299036.04 | .000 | – |
| The model with cross-loadings (prior variances = 0.1) | 4076.99 | 4344.89 | 1263778.32 | 1272267.13 | .000 | .000 |
| The model with cross-loadings (prior variances = 0.1) and residual correlations ($d$ = 200) | -153.61 | 130.49 | 1263325.63 | 1278197.47 | .565 | 1.00 |
| Sample C: Spanish ($n$ = 792) | | | | | | |
| The model with no informative priors | 4881.34 | 5105.61 | 102569.10 | 103346.51 | .000 | – |
| The model with cross-loadings (prior variances = 0.1) | 958.39 | 1213.53 | 98912.79 | 102212.55 | .000 | .000 |
| The model with cross-loadings (prior variances = 0.1) and residual correlations ($d$ = 200) | -85.97 | 204.83 | 98433.42 | 109068.55 | .206 | .998 |
| Sample D: Japanese ($n$ = 549) | | | | | | |
| The model with no informative priors | 2668.25 | 2886.40 | 70756.81 | 71472.86 | .000 | – |
| The model with cross-loadings (prior variances = 0.1) | 824.22 | 1086.16 | 69163.35 | 72323.71 | .000 | .083 |
| The model with cross-loadings (prior variances = 0.1) and residual correlations ($d$ = 200) | -114.50 | 171.73 | 68727.93 | 78883.19 | .345 | .935 |

*Note*. PP$p$: Posterior predictive $p$-value, PPP$p$: Prior-posterior predictive $p$-value

Next, we looked at newly estimated parameters in BSEM with cross-loadings and residual correlations, to explore what makes the model fit of CFA worse. The results are summarized in Table 5. They suggested that most cross-loadings and residual correlations were substantively small. Indeed, PPP$p$ was more than .05 in all samples, suggesting that the hypothesis that a set of parameters are approximately zero was not rejected (Table 3). Thus, the BSEM analysis suggests that minor cross-loadings and residual correlations contributed to the CFA model misfit.

## Discussion

This study aims to clarify the higher-order structure of the PEC with four different samples (French-speaking Belgian, Dutch-speaking Belgian, Spanish, and Japanese). Dimensionality analyses and CFA with MLR revealed that the modified target-based structure (distinction based on the intrapersonal and interpersonal factors with residual correlations among types of competence) fits best among the possible factor structure models, in all samples. This finding emphasizes the importance of distinguishing between intrapersonal and interpersonal domains in EC. Moreover, the results of BSEM showed that model misfit within the modified target-based structure was caused by minor cross-loadings and residual correlations. Given that the strict constraints of exact-zero cross-loadings and residual correlations are unnecessary in the CFA model [38, 46], these results offer further evidence of the validity of the modified target-based structure.

The importance of distinguishing between intrapersonal and interpersonal domains is consistent with theory in EC-related research areas and other fields in psychology. For example, in the related field of emotion regulation, researchers recently developed a theoretical model assuming that perceiving, understanding, and regulating *others*' emotions are related but distinct psychological processes from perceiving, understanding, and regulating one's *own* emotions [68–70]. More broadly, Leary, Raimi [71] indicated the importance of distinguishing intrapersonal from interpersonal motives in a wide range of psychological phenomena, such as cognitive dissonance, biases in decision-making, and self-conscious emotions. The distinction

**Table 4. Results of Bayesian structural equation modeling of the modified target-based structure model (*d* = 200).**

| | Sample A: French-speaking Belgian (*n* = 3295) | | Sample B: Dutch-speaking Belgian (*n* = 9955) | | Sample C: Spanish (*n* = 792) | | Sample D: Japanese (*n* = 549) | |
|---|---|---|---|---|---|---|---|---|
| | Intrapersonal EC | Interpersonal EC | Intrapersonal EC | Interpersonal EC | Intrapersonal EC | Interpersonal EC | Intrapersonal EC | Interpersonal EC |
| Factor loadings | | | | | | | | |
| Identification-self | .96* [.75, 1.19] | -.03 [-.35, .26] | .93* [.71, 1.20] | .02 [-.31, .30] | .99* [.83, 1.20] | -.03 [-.34, .22] | .98* [.80, 1.22] | -.07 [-.40, .19] |
| Comprehension-self | .90* [.73, 1.08] | -.04 [-.29, .18] | .88* [.72, 1.07] | -.03 [-.27, .18] | .90* [.77, 1.07] | -.10 [-.33, .11] | .89* [.72, 1.08] | -.06 [-.31, .17] |
| Expression-self | .69* [.48, .91] | .13 [-.15, .37] | .70* [.46, .94] | .11 [-.18, .36] | .65* [.42, .87] | .18 [-.10, .41] | .67* [.39, .96] | .20 [-.16, .49] |
| Regulation-self | .69* [.50, .86] | -.05 [-.26, .15] | .74* [.51, .94] | -.07 [-.29, .17] | .61* [.39, .79] | -.03 [-.23, .17] | .58* [.35, .79] | .09 [-.15, .30] |
| Utilization-self | .14 [-.13, .39] | .25 [.03, .44] | .14 [-.23, .49] | .23 [-.06, .48] | .20 [-.09, .47] | .29 [.06, .48] | .30 [-.04, .60] | .14 [-.14, .39] |
| Identification-other | .02 [-.21, .23] | .90* [.74, 1.07] | -.03 [-.31, .19] | .96* [.80, 1.19] | .06 [-.19, .29] | .88* [.71, 1.05] | .02 [-.22, .23] | .83* [.65, 1.01] |
| Comprehension-other | .07 [-.19, .29] | .87* [.69, 1.06] | .02 [-.25, .24] | .93* [.74, 1.14] | .11 [-.16, .35] | .89* [.71, 1.07] | .14 [-.09, .35] | .77* [.57, .95] |
| Expression-other | -.14 [-.40, .09] | .88* [.71, 1.07] | -.07 [-.32, .16] | .90* [.71, 1.09] | -.12 [-.39, .11] | .90* [.74, 1.09] | -.09 [-.35, .15] | .79* [.58, .99] |
| Regulation-other | .03 [-.20, .24] | .85* [.67, 1.01] | .06 [-.17, .26] | .80* [.60, .98] | .00 [-.22, .21] | .83* [.65, .98] | -.07 [-.29, .13] | .95* [.80, 1.11] |
| Utilization-other | .09 [-.14, .30] | .40* [.14, .63] | .10 [-.16, .33] | .40* [.08, .67] | .00 [-.21, .20] | .34* [.07, .57] | .04 [-.26, .31] | .85* [.60, 1.06] |
| Factor correlation | | | | | | | | |
| Intrapersonal EC <-> Interpersonal EC | .73* [.50, .87] | | .80* [.60, .93] | | .67* [.38, .86] | | .73* [.48, .89] | |
| Residual correlation | | | | | | | | |
| Identification-self <-> Identification-other | .10 [-.03, .23] | | .00 [-.14, .14] | | -.32* [-.43, -.19] | | .21* [.07, .33] | |
| Comprehension-self <-> Comprehension-other | .27* [.14, .39] | | .18* [.05, .31] | | -.21* [-.33, -.08] | | .30* [.17, .41] | |
| Expression-self <-> Expression-other | .01 [-.12, .14] | | .07 [-.06, .19] | | .01 [-.12, .13] | | .16* [.03, .29] | |
| Regulation-self <-> Regulation-other | .55* [.45, .64] | | .51* [.40, .60] | | .46* [.35, .56] | | .39* [.28, .50] | |
| Utilization-self <-> Utilization-other | .10 [-.01, .21] | | .11 [-.01, .22] | | .10 [-.02, .21] | | .18* [.05, .31] | |

*Note.* 95% credible intervals are in square brackets. EC: emotional competence. Although several upper bounds of 95% credible intervals of standardized factor loadings were higher than one, this is normal and not a problem. For example, the results of Muthén and Asparouhov [46] also show that several upper bounds of 95% credible intervals of standardized factor loadings were higher than one (see https://www.statmodel.com/BSEM.shtml for the their results on credible intervals).

*95% credible interval does not include zero

**Table 5. Frequency distribution of the strength of cross-loadings and residual correlations in the model with cross-loadings (prior variances = 0.1) and residual correlations (*d* = 200).**

| Cross-loadings | $|\beta| < .10$ | $.10 \leq |\beta| < .20$ | $.20 \leq |\beta| < .30$ | $|\beta| \geq .30$ |
|---|---|---|---|---|
| Sample A: French-speaking Belgian | 447 (97.17%) | 11 (2.39%) | 2 (0.44%) | 0 (0.00%) |
| Sample B: Dutch-speaking Belgian | 451 (98.04%) | 6 (1.30%) | 3 (0.65%) | 0 (0.00%) |
| Sample C: Spanish | 449 (97.61%) | 7 (1.52%) | 2 (0.44%) | 2 (0.44%) |
| Sample D: Japanese | 454 (98.70%) | 5 (1.09%) | 1 (0.22%) | 0 (0.00%) |
| Residual correlations | $|r| < .10$ | $.10 \leq |r| < .20$ | $.20 \leq |r| < .30$ | $|r| \geq .30$ |
| Sample A: French-speaking Belgian | 1156 (91.38%) | 102 (8.06%) | 6 (0.47%) | 1 (0.08%) |
| Sample B: Dutch-speaking Belgian | 1115 (88.14%) | 141 (11.15%) | 9 (0.71%) | 0 (0.00%) |
| Sample C: Spanish | 1157 (83.56%) | 188 (14.86%) | 19 (1.50%) | 1 (0.08%) |
| Sample D: Japanese | 1137 (89.88%) | 115 (9.09%) | 10 (0.79%) | 3 (0.24%) |

between intrapersonal and interpersonal domains is increasingly considered as essential to properly understand psychological phenomena.

In the domain of EC, the intrapersonal versus interpersonal higher-order dimensions do more than just provide a parsimonious summary of a complex construct. They are also useful to accurately predict external variables. In fact, previous studies found that intrapersonal and interpersonal EC were differently related to external criteria—for example, intrapersonal EC was more strongly related to objective indices of health [30], depression [33] and regulation of one's own emotions [36, 72], whereas interpersonal EC was more strongly related to behaviors aimed at regulating others' negative emotions [7, 73]. These results suggest that intrapersonal versus interpersonal dimensions can afford more nuanced exploration of relationships between EC and external variables and increase its predictive power.

This study has also implications for emotional education. Emotional education refers to an intervention program aimed at improving EC [74]. Recent research showed that relatively short intervention programs can improve trait-level EC [3, 4]. For effective emotional education, implementers should successfully grasp participants' current level of EC and respond to it. To achieve this goal, the intrapersonal versus interpersonal EC dimensions will be useful to analyze the characteristics of participants' EC profiles and design tailored intervention to foster it. Recent research has strongly called for theory-based EC intervention program that is designed according to a theoretical model of EC [74, 75]. The current results suggest that intrapersonal versus interpersonal dimensions can contribute to this line of research by better differentiating among individuals with different EC profiles and providing a useful framework for designing better emotional education content.

The present study revealed that competence-based factors should be replaced by residual correlations. Nevertheless, among competences, residual correlations between *regulation-self* and *regulation-other* were significant and large after controlling for intrapersonal and interpersonal factors in all samples. This may reflect the fact that individual differences in regulation of one's own emotions are positively associated with those in regulation of another person's emotions. For example, Niven, Totterdell [76] revealed that individual differences in intrinsic affect-improving (the extent to which an individual typically engages in up-regulation of their own emotions) and extrinsic affect-improving (the extent to which that individual typically engages in up-regulation of another person's emotions) were differentiated but positively associated with each other. Such a positive relationship may be represented as significant residual correlation between *regulation-self* and *regulation-other* in the modified target-based structure.

We also found that *utilization-self* did not significantly load on intrapersonal EC in BSEM results, unexpectedly. Several previous studies have also found that facilitating thought using emotions—which is a competence related to utilizing one's own emotions—does not reliably emerge in the factor analysis and is not conceptually distinct from the other competences [77]. For example, factor loadings of the facilitating thought using emotions branch were negligible and not statistically significant beyond the general factor [78]. As discussed in Mayer, Caruso [79], this may be because people utilize their emotions by their emotion comprehension competence (or another competence) rather than any competence distinctly related to facilitating thought. More research is needed to confirm the position of *utilization-self* in EC.

Alongside its strengths, several limitations of this study have to be acknowledged. First, our results are based on self-report measures of EC. Although self-reports are the most widely used method to measure traits and although they have shown evidence of both theoretical and empirical validity [8, 27, 44], traits—including trait-level EC—can also be assessed through observer ratings [80]. Future research should investigate whether the current results can be generalized to alternative methods. Second, given that construct validation is an ongoing

process [81], future research should gather further construct validity evidence such as convergent, discriminant, and predictive validity of the PEC.

Despite these limitations, these findings show the importance of distinguishing between intrapersonal and interpersonal domains in EC. This insight sheds new light on the factor structure of the PEC and opens exciting perspectives for future research.

## Supporting information

**S1 Table. Results of the CFA with a robust maximum likelihood estimation of the competence-based structure model.**
(PDF)

**S2 Table. Results of the CFA with a robust maximum likelihood estimation of the hybrid structure model.**
(PDF)

**S3 Table. Results of the Bayesian structural equation modeling of the modified two second-order factor model with cross-loadings and residual correlations.**
(PDF)

**S1 Text. Details and previous research applications of the hybrid structure model.**
(PDF)

## Acknowledgments

We thank Hervé Avalosse, Rebecca Verniest and Sigrid Vancoreland from the R&D Department from the Mutualité Chrétienne-Christelijke Mutualiteit for their help in the data collection.

## Author Contributions

**Conceptualization:** Yuki Nozaki, Moïra Mikolajczak.

**Data curation:** Yuki Nozaki, Alicia Puente-Martínez, Moïra Mikolajczak.

**Formal analysis:** Yuki Nozaki.

**Funding acquisition:** Yuki Nozaki, Alicia Puente-Martínez.

**Investigation:** Yuki Nozaki, Alicia Puente-Martínez, Moïra Mikolajczak.

**Methodology:** Yuki Nozaki, Alicia Puente-Martínez, Moïra Mikolajczak.

**Supervision:** Moïra Mikolajczak.

**Visualization:** Yuki Nozaki.

**Writing – original draft:** Yuki Nozaki.

**Writing – review & editing:** Yuki Nozaki, Alicia Puente-Martínez, Moïra Mikolajczak.

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
