## [Decision Letter · Decision Letter 0]

6 Aug 2019

PONE-D-19-15440

Evaluating the higher-order structure of emotional competence using the Profile of Emotional Competence (PEC): Confirmatory factor analysis and Bayesian structural equation modeling

PLOS ONE

Dear Dr. Nozaki,

Thank you for submitting your manuscript to PLOS ONE. After careful consideration, we feel that it has merit but does not fully meet PLOS ONE’s publication criteria as it currently stands. Therefore, we invite you to submit a revised version of the manuscript that addresses the points raised during the review process.

Please try to address each of the issues commented by both reviewers. I feel the manuscript may be remarkably improved if done like this.

We would appreciate receiving your revised manuscript by Sep 20 2019 11:59PM. To enhance the reproducibility of your results, we recommend that if applicable you deposit your laboratory protocols in protocols.io, where a protocol can be assigned its own identifier (DOI) such that it can be cited independently in the future. For instructions see: http://journals.plos.org/plosone/s/submission-guidelines#loc-laboratory-protocols

We look forward to receiving your revised manuscript.

Kind regards,

Juan Carlos Perez-Gonzalez, Ph.D.

Academic Editor

PLOS ONE

Journal Requirements:

2. Please provide additional details regarding participant consent. In the Methods section, please ensure that you have specified (1) whether consent was informed and (2) what type you obtained (for instance, written or verbal). If your study included minors, state whether you obtained consent from parents or guardians. If the need for consent was waived by the ethics committee, please include this information.

Reviewers' comments:

Reviewer's Responses to Questions

**Comments to the Author**

1. Is the manuscript technically sound, and do the data support the conclusions?

Reviewer #1: Partly

Reviewer #2: Yes

2. Has the statistical analysis been performed appropriately and rigorously? 

Reviewer #1: Yes

Reviewer #2: Yes

3. Have the authors made all data underlying the findings in their manuscript fully available?

Reviewer #1: No

Reviewer #2: Yes

4. Is the manuscript presented in an intelligible fashion and written in standard English?

Reviewer #1: Yes

Reviewer #2: Yes

5. Review Comments to the Author

Reviewer #1: Thank-you for the opportunity to review your manuscript. I approach this review from the perspective of an Emotional Intelligence researcher, with experience testing models of factor structure of EI constructs using MPlus (but not Bayesian SEM). I am very passionate about the field so was thrilled to have the opportunity to review your work, and I do hope my comments are of benefit to you for this project.

To explain my responses to the set questions, some of the code and data are in a private OSF page which I cannot currently access. As for Q1, please see comments below regarding scale vs construct.

My major concern with this manuscript is that there is a little conflation of measure and construct. The theoretical clarity of EI has often been questioned, and so when you suggest EI and EC are interchangeable terms on p2/3 I think this has more scope for causing confusion than clarity. For example, some researchers differentiate between cognitive ability (emotional intelligence), trait (affect-related personality) and behaviour (e.g. emotion regulation). It could be clearer exactly how you expect the competency perspective to fit in/around these other domains. This concern is furthered by the critique of competencies as atheoretical/ambiguous both within the EI domain, and much broader. The lack of theoretical position on competencies means there are some areas of ambiguity throughout the manuscript. For example, why would dimensions be excluded for not representing both ability and trait approaches – why would this matter if they were something different to these perspectives, or only related to one? As a result of these sorts of concerns, I think this paper would be much more valuable if it represented a discussion on the structure of the scale, rather than claiming for a definitive model of competencies.

Your detailing, presentation and analysis of the models looks pretty clear although some further details would be of benefit. For example, the justification for MLR over WLSMV is only convincing if you make the argument that the indicators provide additional value in model discrimination. I don’t have any experience with BSEM, although I am familiar with the principles, and so found some of the analysis section a little tricky to follow. My main problem with this is that it can be easy to miss errors or make subjective decisions which are difficult to follow. As such, I believe the analyses are robust but some of the more technical notes could be made a little easier to follow.

Your discussion is mostly clear, although I have two comments here. Firstly, the speculation for previous findings on lines 365-370 I would not encourage. Secondly, the fact that utilization-self did not load in the BSEM results is not a limitation of the current research- this finding should be discussed in the results or discussion section more substantively. You make an interesting line of argument around it not being include in factor structures of ability (e.g. see also Fan et al.) however again this links back to my first substantive comment about theoretical ambiguity.

As a whole, I really enjoyed reading this manuscript and found it in the most part to represent clear communication of a substantive contribution to the field. However, due to the issues around theory, I would encourage this paper to be re-orientated towards the measure rather than the construct. I do hope this is all clear and helps, I am very happy to elaborate wherever of value. Thank-you for the opportunity to read your work!

Reviewer #2: Manuscript PONE –D-19-15440

Full title: Evaluating the higher-order structure of emotional competence using the Profile of Emotional Competence (PEC): Confirmatory factor analysis and Bayesian structural equation modeling.

The study represents an important attempt to integrate previous findings into a model of Emotional Competences that goes beyond the traditional emotional intelligence paradigms. The authors utilize advanced statistical techniques to demonstrate that the five factor structure of Emotional Competences is relatively stable across cultures and show that the model distinguishing between intrapersonal and interpersonal domains fit the data best. Findings from this study can assist researchers and EI program developers who would like to base their implementations on a solid and evidence-based model of EC.

There is tremendous value in investigating systematically the higher order structure of Emotional Competence in different cultures, and findings from this study would be very helpful for the design of future EI implementation programs on EI.

The authors successfully introduce a complex theoretical distinction and clarify terminology issues, which have been a problem during decades of emotional intelligence research. The use of very relevant and up-to-date references is noticeable.

At the end of the description of the PEC the authors refer to the fact that the modification of the Likert scale for one of their samples did not affect the main results. It would be helpful if they could clarify this sentence and indicate how they reached this conclusion. In addition, it would be advisable to incorporate examples of items in the PEC illustrating every component of the test.

Some demographic variables seem to be quite different across samples (e.g., main age), as well as the questionnaire administration conditions (through data collection company online for the Japanese data only), so it would be advisable to identify clearly how the authors dealt with these differences.

The authors made great efforts in interpreting the results regarding interpersonal and intrapersonal EC. In order to further enhance this discussion, the authors could refer to how these findings could inform future implementation programs aiming to develop EC in particular, and Emotional Education in general (see Perez-Gonzalez & Qualter, 2018).

As for the recommendations for future research, specific comments on future validation of the model could be made (content validity, discriminatory power, etc.). Also, the authors could elaborate on previous validation work using the PEC earlier on in the manuscript.

Overall, this is a coherent and solid research paper, which adds to the existent literature on emotional intelligence. I definitely recommend its publication in PlosOne

6. PLOS authors have the option to publish the peer review history of their article (what does this mean?). If published, this will include your full peer review and any attached files.

Reviewer #1: No

Reviewer #2: Yes: Maria-Jose Sanchez-Ruiz

---

## [Author Response · Author response to Decision Letter 0]

28 Aug 2019

We would like to express our sincere gratitude to the editor and the reviewer for their insightful comments. Their comments help us to improve the paper significantly.

Responses to the Editor

Comment 1: Please ensure that your manuscript meets PLOS ONE's style requirements, including those for file naming. 

Response: We have rechecked the guideline and ensured that our manuscript meets PLoS ONE’s style requirements.

Comment 2: Please provide additional details regarding participant consent. In the Methods section, please ensure that you have specified (1) whether consent was informed and (2) what type you obtained (for instance, written or verbal). If your study included minors, state whether you obtained consent from parents or guardians. If the need for consent was waived by the ethics committee, please include this information.

Response: In accordance with the comment, we have specified participant consent in the revised manuscript (lines 142–146).

Responses to Reviewer 1

Comment 1: To explain my responses to the set questions, some of the code and data are in a private OSF page which I cannot currently access.

Response: We apologize for this error. We have corrected it as suggested. 

Comment 2: My major concern with this manuscript is that there is a little conflation of measure and construct. The theoretical clarity of EI has often been questioned, and so when you suggest EI and EC are interchangeable terms on p2/3 I think this has more scope for causing confusion than clarity. For example, some researchers differentiate between cognitive ability (emotional intelligence), trait (affect-related personality) and behaviour (e.g. emotion regulation). It could be clearer exactly how you expect the competency perspective to fit in/around these other domains. This concern is furthered by the critique of competencies as atheoretical/ambiguous both within the EI domain, and much broader. The lack of theoretical position on competencies means there are some areas of ambiguity throughout the manuscript. For example, why would dimensions be excluded for not representing both ability and trait approaches – why would this matter if they were something different to these perspectives, or only related to one? As a result of these sorts of concerns, I think this paper would be much more valuable if it represented a discussion on the structure of the scale, rather than claiming for a definitive model of competencies.

Response: Thank you for this thoughtful suggestion. As we explained in the manuscript, we use the term “emotional competence” rather than “emotional intelligence” because it can be improved via relatively short trainings and we operationalized the construct through the PEC. This is consistent with previous research using the PEC (e.g., Batselé, Stefaniak, & Fantini-Hauwel, 2019; Nozaki & Koyasu, 2016; Szczygiel & Mikolajczak, 2019). However, we agree that we should include a discussion on the structure of the scale, rather than claiming for a definitive model of competencies.

Thus, we have revised the Introduction section to be oriented toward the scale (i.e., Profile of Emotional Competence; PEC), rather than the construct (lines 43–125). 

Comment 3: Your detailing, presentation and analysis of the models looks pretty clear although some further details would be of benefit. For example, the justification for MLR over WLSMV is only convincing if you make the argument that the indicators provide additional value in model discrimination. I don’t have any experience with BSEM, although I am familiar with the principles, and so found some of the analysis section a little tricky to follow. My main problem with this is that it can be easy to miss errors or make subjective decisions which are difficult to follow. As such, I believe the analyses are robust but some of the more technical notes could be made a little easier to follow.

Response: We have added an explanation that AIC and BIC are frequently used for model comparison (lines 181–183). Moreover, we have rechecked the contents of the analysis section while referring to reported information in previous research using BSEM (e.g., de Beer & Bianchi, in press).

Comment 4: Your discussion is mostly clear, although I have two comments here. Firstly, the speculation for previous findings on lines 365-370 I would not encourage. Secondly, the fact that utilization-self did not load in the BSEM results is not a limitation of the current research- this finding should be discussed in the results or discussion section more substantively. You make an interesting line of argument around it not being include in factor structures of ability (e.g. see also Fan et al.) however again this links back to my first substantive comment about theoretical ambiguity.

Response: In accordance with the comment, we have deleted lines 365–370 in the previous manuscript. Moreover, we have moved the discussion about utilization-self from the limitation section to the discussion section (lines 380–387).

Thank you again for your helpful and encouraging comments on the manuscript. We very much hope that you will find the revised manuscript suitable for publication.

Responses to Reviewer 2

Comment 1: The study represents an important attempt to integrate previous findings into a model of Emotional Competences that goes beyond the traditional emotional intelligence paradigms. The authors utilize advanced statistical techniques to demonstrate that the five factor structure of Emotional Competences is relatively stable across cultures and show that the model distinguishing between intrapersonal and interpersonal domains fit the data best. Findings from this study can assist researchers and EI program developers who would like to base their implementations on a solid and evidence-based model of EC. There is tremendous value in investigating systematically the higher order structure of Emotional Competence in different cultures, and findings from this study would be very helpful for the design of future EI implementation programs on EI. The authors successfully introduce a complex theoretical distinction and clarify terminology issues, which have been a problem during decades of emotional intelligence research. The use of very relevant and up-to-date references is noticeable.

Response: Thank you for your comment. We are grateful for your acknowledgment of the value of our research.

Comment 2: At the end of the description of the PEC the authors refer to the fact that the modification of the Likert scale for one of their samples did not affect the main results. It would be helpful if they could clarify this sentence and indicate how they reached this conclusion. In addition, it would be advisable to incorporate examples of items in the PEC illustrating every component of the test.

Response: We found similar factor structure across all samples (i.e., the modified target-based structure is best). Based on this result, we reached this conclusion. We have added this explanation in the revised manuscript (lines 165–167. Moreover, we have added example items of the PEC in the revised manuscript (lines 150–160). 

Comment 3: Some demographic variables seem to be quite different across samples (e.g., main age), as well as the questionnaire administration conditions (through data collection company online for the Japanese data only), so it would be advisable to identify clearly how the authors dealt with these differences.

Response: As we explained in the manuscript, the main purpose of using four different samples is to test the stability and replicability of the results. Given that factor structure is usually same across different age or gender groups (e.g., Tsaousis & Kazi, 2013), we expect this difference will not significantly affect our results. We just tested whether the results were stable and replicable across samples and found that the modified target-based structure is best in all samples. Thus, no special method is need to dealt with differences in demographic variables in this study.

Furthermore, participants in all samples answered the questionnaire online. Thus, the questionnaire administration condition was similar across samples. We have clarified this point in the revised manuscript (line 142).

Comment 4: The authors made great efforts in interpreting the results regarding interpersonal and intrapersonal EC. In order to further enhance this discussion, the authors could refer to how these findings could inform future implementation programs aiming to develop EC in particular, and Emotional Education in general (see Perez-Gonzalez & Qualter, 2018).

Response: In accordance with the comment, we have added implication for EC training programs in the discussion section (line 358–367).

Comment 5: As for the recommendations for future research, specific comments on future validation of the model could be made (content validity, discriminatory power, etc.). Also, the authors could elaborate on previous validation work using the PEC earlier on in the manuscript.

Response: In accordance with the comment, we have suggested the need for future validation work in the discussion section (lines 393–395). Moreover, we have cited previous validation works that are using the PEC in the Introduction section (line 52–53).

Thank you again for your helpful comments on the manuscript. We hope that you will find the revised manuscript suitable for publication.

---

## [Decision Letter · Decision Letter 1]

4 Oct 2019

PONE-D-19-15440R1

Evaluating the higher-order structure of emotional competence using the Profile of Emotional Competence (PEC): Confirmatory factor analysis and Bayesian structural equation modeling

PLOS ONE

Dear Dr. Nozaki,

Thank you for submitting your manuscript to PLOS ONE. After careful consideration, we feel that it has merit but does not fully meet PLOS ONE’s publication criteria as it currently stands. Therefore, we invite you to submit a revised version of the manuscript that addresses the points raised during the review process for one of the two reviewers.

As Academic Editor, this time I feel the required minor changes will bring the manuscript direct to final acceptance if you follow the reviewer' guidelines. I frankly agree with the reviewer's observations and I believe that his observations can significantly strengthen the theoretical robustness of the paper, providing a necessary refinement.

We would appreciate receiving your revised manuscript by Nov 18 2019 11:59PM. To enhance the reproducibility of your results, we recommend that if applicable you deposit your laboratory protocols in protocols.io, where a protocol can be assigned its own identifier (DOI) such that it can be cited independently in the future. For instructions see: http://journals.plos.org/plosone/s/submission-guidelines#loc-laboratory-protocols

We look forward to receiving your revised manuscript.

Kind regards,

Juan-Carlos Perez-Gonzalez, Ph.D.

Academic Editor

PLOS ONE

Reviewers' comments:

Reviewer's Responses to Questions

**Comments to the Author**

1. If the authors have adequately addressed your comments raised in a previous round of review and you feel that this manuscript is now acceptable for publication, you may indicate that here to bypass the “Comments to the Author” section, enter your conflict of interest statement in the “Confidential to Editor” section, and submit your "Accept" recommendation.

Reviewer #1: (No Response)

Reviewer #2: All comments have been addressed

2. Is the manuscript technically sound, and do the data support the conclusions?

Reviewer #1: Yes

Reviewer #2: Yes

3. Has the statistical analysis been performed appropriately and rigorously? 

Reviewer #1: Yes

Reviewer #2: Yes

4. Have the authors made all data underlying the findings in their manuscript fully available?

Reviewer #1: Yes

Reviewer #2: Yes

5. Is the manuscript presented in an intelligible fashion and written in standard English?

Reviewer #1: Yes

Reviewer #2: Yes

6. Review Comments to the Author

Reviewer #1: Thank-you for the opportunity to re-review this work. I believe this work is of high-quality and makes an interesting contribution to the field. Many of my comments have been addressed and the paper as a whole reads very well. Excluding a few minor comments highlighted below, my main concerns surrounding the framing of the introduction remain.

You have done a good job at reframing the work to be about the scale rather than the construct, however your introduction still presents a slightly confused review of EI theory. For example, you conflate EC with trait EI (p3) – particularly in the second paragraph (starting “Research conducted over the last two decades…”) where you cite mostly trait EI research, however some may argue that your central argument that EC is malleable to training is much less true for trait EI. Furthermore, this becomes slightly further confused when suggesting the PEC is built upon the Mayer and Salovey model (p4) which itself is an ability EI framework. I would recommend you take a clearer stance on this – this could be done in a number of different ways – you could argue that EC is different to ability EI and trait EI, you could argue that trait EI and EC are in essence the same, you could argue that competencies are the behavioural outcomes of ability EI (hence the Mayer and Salovey framework) and trait EI (hence self-report behavioural measure), or take it in a different way. Whichever way, framing it more clearly and acknowledging the consequences of this decision is vital. As it stands this is the major obstacle to publication for this work is this theoretical clarity which frames the whole paper.

Minor changes/recommendations/thoughts:

• Remove ‘emotional competence using’ from the title

• P3. Trait EI… is TYPICALLY measured using self-report. It’s worth adding this conditional statement as there has been a body of research considering other-rated trait EI/competencies

• P13 there are two ‘that’ in the sentence – that that

• P13 your lack of convergence sounds like a Heywood case. Is that the case and if so did you attempt any resolution?

• P26 you link the findings to emotional education as requested by the second reviewer – I would encourage you to expand on this just a little more in context of the theoretical approach adopted. I.e. if you take a trait or competency approach, what is the evidence that such individual differences can be trained, and what would you expect to improve from understanding the factor structure differentiating between interpersonal and intrapersonal?

• P27 you discuss the utilisation branch and acknowledge in other factor analytic work that this does not emerge. Could you provide just a little more detail about your findings in this area i.e. do loadings or relationships to other factors seem higher than the others etc.

In sum, there is some great work presented and I fully support the publication of this work, but the theoretical grounding to the constructs discussed needs refinement. Other (minor) recommendations and thoughts are provided to further refine the manuscript. I do hope these help!

Reviewer #2: The authors have successfully answered all the comments raised and I recommend this manuscript for publication in PLOSONE.

7. PLOS authors have the option to publish the peer review history of their article (what does this mean?). If published, this will include your full peer review and any attached files.

Reviewer #1: No

Reviewer #2: Yes: Maria-Jose Sanchez-Ruiz

---

## [Author Response · Author response to Decision Letter 1]

19 Oct 2019

We would like to express our sincere gratitude to the editor and the reviewer for their insightful comments. Their comments help us to improve the paper significantly.

Responses to Reviewer 1

Comment 1: You have done a good job at reframing the work to be about the scale rather than the construct, however your introduction still presents a slightly confused review of EI theory. For example, you conflate EC with trait EI (p3) – particularly in the second paragraph (starting “Research conducted over the last two decades…”) where you cite mostly trait EI research, however some may argue that your central argument that EC is malleable to training is much less true for trait EI. Furthermore, this becomes slightly further confused when suggesting the PEC is built upon the

Mayer and Salovey model (p4) which itself is an ability EI framework. I would recommend you take a clearer stance on this – this could be done in a number of different ways – you could argue that EC is different to ability EI and trait EI, you could argue that trait EI and EC are in essence the same, you could argue that competencies are the behavioural outcomes of ability EI (hence the Mayer and Salovey framework) and trait EI (hence self-report behavioural measure), or take it in a different way. Whichever way, framing it more clearly and acknowledging the consequences of this decision is vital. As it stands this is the major obstacle to publication for this work is this theoretical clarity which frames the whole paper.

Response: A recent meta-analysis shows that relatively short trainings can improve all EI based on the ability model, the trait model, and the mixed model, unlike cognitive intelligence (Hodzic et al., 2018). Given this line of research, we think emotional competence (EC) is a more suitable term to refer to these individual differences than emotional intelligence (EI). This is why we use the term EC instead of EI in the manuscript. We have added this explanation in the main text (lines 22–27). Moreover, to increase the theoretical clarity, we have also added the explanation of the tripartite model proposed by Mikolajczak et al. (2009) to clarify how different levels of EC/EI (knowledge, abilities and traits) are related to each other (lines 31–40) and what the terms “EC” used here refers to.

Comment 2: Remove ‘emotional competence using’ from the title

Response: In accordance with the comment, we have removed them from the title.

Comment 3: P3. Trait EI… is TYPICALLY measured using self-report. It’s worth adding this conditional statement as there has been a body of research considering other-rated trait EI/competencies

Response: In accordance with the comment, we have added “typically” to that sentence (line 41).

Comment 4: P13 there are two ‘that’ in the sentence – that that

Response: We have fixed this mistake (lines 236).

Comment 5: P13 your lack of convergence sounds like a Heywood case. Is that the case and if so did you attempt any resolution?

Response: With regards to the competence-based structure and modified competence-based structure, it is a Heywood case, because some correlation coefficients among second-order factors exceeded 1.00 (see S1 Table). We might try to fix this type of issue by using inequality constraint options (set correlations < 1.00) during estimation). However, this procedure is not recommended in the previous literature because these ‘‘fixes’’ obscure important sources of model misspecification (Fan & Lance, 2017; Kline, 2010). Rather, previous literature suggests that we should try to inspect the source of the problem and interpret them (Kline, 2010). As the S1 Table clearly shows, many correlation coefficients among second-order factors exceeded 1.00. Thus, instead of unreasonable attempts to fix these issues, we interpreted the source of the improper solution and conclude that the factors were overextracted in the competence-based structure and modified competence-based structure.

Comment 6: P26 you link the findings to emotional education as requested by the second reviewer – I would encourage you to expand on this just a little more in context of the theoretical approach adopted. i.e. if you take a trait or competency approach, what is the evidence that such individual differences can be trained, and what would you expect to improve from understanding the factor structure differentiating between interpersonal and intrapersonal?

Response: We have added recent findings showing that trait-level EC can be improved through training programs and discussed the value of intra- and interpersonal dimensions for emotional education (lines 365–375).

Comment 7: P27 you discuss the utilisation branch and acknowledge in other factor analytic work that this does not emerge. Could you provide just a little more detail about your findings in this area i.e. do loadings or relationships to other factors seem higher than the others etc.

Response: We have added this information to the revised manuscript (lines 392–393).

Thank you again for your helpful and encouraging comments on the manuscript. We very much hope that you will find the revised manuscript suitable for publication.

Responses to Reviewer 2

Comment 1: The authors have successfully answered all the comments raised and I recommend this manuscript for publication in PLOSONE.

Response: Thank you for your comment. We are grateful for your acknowledgment of the value of our research.

---

## [Editor Report · Decision Letter 2]

29 Oct 2019

Evaluating the higher-order structure of the Profile of Emotional Competence (PEC): Confirmatory factor analysis and Bayesian structural equation modeling

PONE-D-19-15440R2

Dear Dr. Nozaki,

Great work! We are pleased to inform you that your manuscript has been judged scientifically suitable for publication and will be formally accepted for publication once it complies with all outstanding technical requirements.

With kind regards,

Juan-Carlos Pérez-González, Ph.D.

Academic Editor

PLOS ONE
---

## [Editor Report · Acceptance letter]

5 Nov 2019

PONE-D-19-15440R2 

Evaluating the higher-order structure of the Profile of Emotional Competence (PEC): Confirmatory factor analysis and Bayesian structural equation modeling 

Dear Dr. Nozaki:

I am pleased to inform you that your manuscript has been deemed suitable for publication in PLOS ONE. Congratulations! Your manuscript is now with our production department. 

With kind regards,

on behalf of

Dr. Juan-Carlos Pérez-González 

Academic Editor

PLOS ONE